# AMPKα1 Regulates Lung and Breast Cancer Progression by Regulating TLR4-Mediated TRAF6-BECN1 Signaling Axis

**DOI:** 10.3390/cancers12113289

**Published:** 2020-11-06

**Authors:** Mi-Jeong Kim, Yoon Min, Juhee Son, Ji Young Kim, Ji Su Lee, Duk-Hwan Kim, Ki-Young Lee

**Affiliations:** 1Department of Immunology, Sungkyunkwan University School of Medicine, 2066 Seobu-ro, Jangan-gu, Suwon, Gyeonggi-do 16419, Korea; kmjj0107@skku.edu (M.-J.K.); unizzzang@skku.edu (Y.M.); kimthsh@skku.edu (J.S.); bh9506@nate.com (J.Y.K.); alsdk0376@naver.com (J.S.L.); 2Department of Molecular Cell Biology, Sungkyunkwan University School of Medicine, 2066 Seobu-ro, Jangan-gu, Suwon, Gyeonggi-do 16419, Korea; dukhwan@skku.edu; 3Samsung Biomedical Research Institute, Sungkyunkwan University School of Medicine, 2066 Seobu-ro, Jangan-gu, Suwon, Gyeonggi-do 16419, Korea; 4Samsung Medical Center, Seoul 06351, Korea; 5Department of Health Sciences and Technology, Samsung Advanced Institute for Health Sciences & Technology, Samsung Medical Center, Sungkyunkwan University, 81 Irwon-ro, Gangnam-gu, Seoul 06351, Korea

**Keywords:** AMPKαl, toll like receptor 4, TRAF6-BECN1 signaling axis, autophagy, cancer

## Abstract

**Simple Summary:**

TRAF6-BECN1 signaling axis in TLR4 signal plays an essential role for the autophagy induction, thereby it regulates cancer migration and invasion. Here we show that *AMPKα1*, one of the isoforms of AMPK, is functionally involved in autophagy induction by regulating the TRAF6-BECN1 signaling axis. In this context, *AMPKα1*-knockout lung or breast cancer cells exhibited the attenuation of cancer cell migration and invasion induced by TLR4 simulation. Additionally, we could find that the expression of AMPKα1 is positively associated with gene expressions related to autophagy, migration, and metastasis of cancer cells in primary non-small cell lung cancers (NSCLCs). These findings demonstrate that AMPKα1 plays a pivotal role in cancer progression by regulating the TRAF6-BECN1 signaling axis for autophagy induction.

**Abstract:**

TRAF6-BECN1 signaling axis is critical for autophagy induction and functionally implicated in cancer progression. Here, we report that AMP-activated protein kinase alpha 1 (*AMPKα1*, *PRKAA1*) is positively involved in autophagy induction and cancer progression by regulating TRAF6-BECN1 signaling axis. Mechanistically, AMPKα1 interacted with TRAF6 and BECN1. It also enhanced ubiquitination of BECN1 and autophagy induction. *AMPKα1*-knockout (*AMPKα1*KO) HEK293T or *AMPKα1*-knockdown (*AMPKα1*KD) THP-1 cells showed impaired autophagy induced by serum starvation or TLR4 (Toll-like receptor 4) stimulation. Additionally, *AMPKα1*KD THP-1 cells showed decreases of autophagy-related and autophagosome-related genes induced by TLR4. *AMPKα1*KO A549 cells exhibited attenuation of cancer migration and invasion induced by TLR4. Moreover, primary non-small cell lung cancers (NSCLCs, *n* = 6) with low AMPKαl levels showed markedly decreased expression of genes related to autophagy, cell migration and adhesion/metastasis, inflammation, and TLRs whereas these genes were significantly upregulated in NSCLCs (*n* = 5) with high AMPKαl levels. Consistently, attenuation of cancer migration and invasion could be observed in *AMPKα1*KO MDA-MB-231 and *AMPKα1*KO MCF-7 human breast cancer cells. These results suggest that AMPKα1 plays a pivotal role in cancer progression by regulating the TRAF6-BECN1 signaling axis for autophagy induction.

## 1. Introduction

AMP-activated protein kinase (AMPK) is a heterotrimeric protein complex composed of α, β, and γ subunits. It plays an essential role in the regulation of cellular metabolism to maintain energy homeostasis [1]. Accumulating evidence has shown that AMPK is functionally implicated in various cellular responses, such as autophagy [2,3], inflammation [4,5], and immunity [6,7,8]. In mammals, autophagy induction is regulated under nutrient-rich and starvation conditions by TOR and AMPK complex [9]. In mammals, ULK1 and ULK2 are founded in a complex with ATG13, FIP200, and ATG101. They are functionally linked to starvation-induced autophagy [9,10,11]. Under a starvation condition, ULK1/2 is activated by AMPK through phosphorylation [10,11]. Simultaneously, AMPK induces phosphorylation of raptor, a regulatory-associated protein of MTOR (mechanistic target of rapamycin kinase), leading to inhibition of MTORC1 (MTOR complex 1) and induction of autophagy [9,10,11,12]. In addition, AMPK induces phosphorylation of Beclin 1 (*BECN1*), an essential protein for the function of class III phosphatidylinositol 3-kinase (PtdIns3K) complexes and TRAF6-BECN1 signaling axis for autophagy induction [10,11,13,14].

Beclin 1 (*BECN1*), one of the first identified mammalian autophagy-related proteins, is a component of PtdIns3K complexes. It plays an essential role in autophagy flux and induction through its ubiquitination and phosphorylation [9,10,11,12]. A recent report has shown that phosphorylation of BECN1 at threonine 388 induced by AMPK is critical for autophagy regulation [11]. AMPK-mediated BECN1 phosphorylation induces the accumulation of LC3 puncta, leading to autophagy induction [11]. Besides BECN1 phosphorylation, TRAF6-induced ubiquitination of BECN1 also plays a key role in TLR-induced autophagy activation [13,14]. Upon TLR stimulation, a lysine located at the Bcl-2 homology 3 (BH3) domain of BECN 1 serves as a major site for K63-linked ubiquitination by TRAF6, leading to regulation of autophagy [13,14]. Importantly, the TRAF6-BECN1 signaling axis regulated by TLRs facilitates both migration and invasion of cancer cells through autophagy induction [13,14,15,16]. These results suggest that molecular regulation of BECN1 phosphorylation and ubiquitination by AMPK and TRAF6, respectively, might play a central role in autophagy induction, thus affecting cancer progression through autophagy induction. 

Autophagy plays a key role in the regulation of cancer progression [17,18]. Accumulating evidence has shown that autophagy promotes cancer progression including migration, invasion, and metastasis of many different cancers by supplying essential nutrients for tumor growth [17,18,19,20]. By recycling intracellular proteins and organelles by autophagy, tumor cells are supplied with a variety of nutrients as a critical energy source during metabolic stress to maintain homeostasis and viability [19,20]. Additionally, it has been demonstrated that autophagy plays a key role in tumor necrosis and metabolic damage in response to cellular stress [19,20]. By environmental stimuli, including nutrient deprivation, growth factor depletion, and hypoxia, cytoplasmic components are degraded, promoting cell survival by activating autophagy [19,20]. Although the function of autophagy in cancer progression is somewhat controversial, current evidence indicates that autophagy is functionally implicated in cancer progression.

To further understand molecular and cellular mechanisms involved in the regulation of autophagy induction by AMPKαl through TRAF6-BECN1 signaling axis, thereby regulating cancer progression, we explored the functional role of AMPKαl in cancer migration and invasion of *AMPKαl*-knockout (*AMPKα1*KO) lung cancer and breast cancer cells in the present study and demonstrated its functional association in *AMPKαl*-knockdown (*AMPKα1*KD) cells, 42 primary non-small cell lung cancers (NSCLCs). Through biochemical and cellular studies, together, we herein provide evidence that AMPKαl can positively regulate the TRAF6-BECN1 signaling axis for autophagy induction, thereby regulating cancer progression.

## 2. Materials and Methods

### 2.1. Study Population 

Surgically resected tumor and matched normal tissues (*n* = 42) were obtained from 42 NSCLC patients who underwent curative surgical resection at the Department of Thoracic and Cardiovascular Surgery, Samsung Medical Center, Seoul, Korea, as previously described [21]. The pathological stage of NSCLC was determined using the tumor/node/metastasis (TNM) system provided by the American Joint Committee on Cancer (AJCC) [22]. This study was conducted in accordance with ethical principles stated in the Declaration of Helsinki and approved by the Institutional Review Board (IRB#: 2010-07-204) of the Samsung Medical Center. Written informed consent to use pathological specimens for research was obtained from each patient prior to surgery. All data are not publicly available due to privacy and ethical restriction.

### 2.2. Cell Lines and Cell Culture

Human embryonic kidney (HEK) 293T cells (ATCC, CRL-11268) were cultured and maintained in Dulbecco’s modified Eagle’s medium (DMEM; Thermo Fisher Scientific, 11965092, Waltham, MA, USA). Human monocytic THP-1 cells (ATCC, TIB-202) were maintained in RPMI (Roswell Park Memorial Institute) 1640 medium (Thermo Fisher Scientific, 11875093, Waltham, MA, USA) supplemented with 10% fetal bovine serum (FBS; Fisher Scientific HyClone, 11306060, Waltham, MA, USA), 2 mM L-glutamine (GIBCO, A2916801, Rockville, MD, USA), 100 units/mL penicillin (GIBCO, 15140122, Rockville, MD, USA), 100 μg/mL streptomycin (GIBCO, 15140122, Rockville, MD, USA), and 5 × 10^−5^ M β-mercaptoethanol (GIBCO, 21985023, Rockville, MD, USA). A549 cells (human lung cancer cell line; ATCC, CCL-185, Manassas, VA, USA), MDA-MB-231 cells (human breast carcinoma cell line; ATCC, HTB-26, Manassas, VA, USA), and MCF-7 cells (human breast carcinoma cell line; ATCC, HTB-22, Manassas, VA, USA) were maintained in RPMI supplemented with 10% FBS.

### 2.3. Antibodies and Reagents

Anti-AMPKαl (Abcam, ab3759, Cambridge, MA, USA), anti-TAK1 (Cell Signaling, #4505, Danvers, MA, USA), anti-pho-TAK1 (Cell Signaling, #4531, Danvers, MA, USA), anti-Myc (Cell Signaling, #2276, Danvers, MA, USA), anti-GAPDH (Cell Signaling, #2118, Danvers, MA, USA), anti-LC3A/B (Cell Signaling, #4108, Danvers, MA, USA), and anti-Flag (Sigma-Aldrich, SAB4200071, St Louis, MO, USA) antibodies were obtained for this study. Lipopolysaccharide (LPS, Sigma-Aldrich, serotype 0128: B12, St Louis, MO, USA), chloroquine (CQ; Sigma-Aldrich, C6628, St Louis, MO, USA), dimethyl sulfoxide (DMSO; Sigma-Aldrich, 472301, St Louis, MO, USA), puromycin (Sigma-Aldrich, P8833, St Louis, MO, USA), paraformaldehyde (Sigma-Aldrich, P6148, St Louis, MO, USA), Triton X-100 (Sigma-Aldrich, T8787, St Louis, MO, USA), 3-Methyladenine (3-MA; Sigma-Aldrich, M9281, St Louis, MO, USA), gentamicin (Sigma-Aldrich, G1272, St Louis, MO, USA), deoxycholate (Sigma-Aldrich, D6750, St Louis, MO, USA), Lipofectamine 2000 (Thermo Scientific, 11668019, Waltham, MA, USA), and Dulbecco’s phosphate-buffered saline (DPBS; Sigma-Aldrich, D8537, St Louis, MO, USA) were purchased and used.

### 2.4. Generation of AMPKα1-Knockdown (AMPKα1KD) THP-1 Cells

To generate control (Ctrl) THP-1 and *AMPKα1*KD THP-1 cells, lentivirus containing small hairpin RNA (shRNA) targeting human AMPKαl (Santa Cruz Biotechnology, sc-29673-V, Santa Cruz, CA, USA) and control shRNA lentivirus (Santa Cruz Biotechnology, Santa Cruz, sc-108080, Santa Cruz, CA, USA) were used to infect THP-1 cells. Ctrl THP-1 and AMPKα1KD THP-1 cells were selected as previously described [8,23,24,25,26].

### 2.5. Generation of AMPKαl-Knockout (AMPKα1KO) Cancer Cell Lines With CRISPR/Cas9

Guide RNA sequences for CRISPR/Cas9 were designed. *AMPKα1*KO cells were generated as previously described [13,15,23]. Insert oligonucleotides for human AMPKαl gRNA were 5′-CACCGGAAGATTCGGAGCCTTGATG-3′/3′-CCTTCTAAGCCTCGGAACTACCAAA-5′. Complementary oligonucleotides for guide RNAs (gRNAs) were annealed and cloned into lentiCRISPR v2 vector (Addgene plasmid, Ca#52961, Cambridge, MA, USA). LentiCRISPR v2/gRNA was transfected into HEK293T, A549, MCF-7, or MDA-MB-231 cells using Lipofectamine 2000 according to the manufacturer’s instructions. *AMPKα1*KO colonies were isolated as previously described and confirmed using western blot [13,15,23].

### 2.6. Microarray Analysis

Ctrl THP-1 and *AMPKα1*KD THP-1 cells were treated with or without LPS for different times. Total RNA was extracted using Trizol (Thermo Fisher Scientific, 15596026, Waltham, MA, USA) and purified using RNeasy columns (Qiagen, 74106, Chatsworth, CA, USA) according to each manufacturers’ protocol. Total RNAs were extracted in tumor and matched normal tissues from 42 NSCLC patients as previously described [21]. Microarray analysis was performed and analyzed as previously described [8,24,27].

### 2.7. Plasmid Constructs

FLAG-TRAF6 (Addgene, 21624, Cambridge, MA, USA) and FLAG-BECN1 (Addgene, 24388, Cambridge, MA, USA) were used in this study. FLAG-AMPKαl and AMPKαl (D159A) DN were kindly provided by Dr. S. H. Um (Sungkyunkwan University School of Medicine, Korea). HA-Ub was obtained from Dr. J. H. Shim (University of Massachusetts Medical School, USA). Truncated mutants of FLAG-TRAF6, FLAG-AMPKαl, and MYC-BECN1 were generated as previously described [8,15,16,23,24,27].

### 2.8. Western Blotting Analysis and Immunoprecipitation (IP) Assays

Western blotting and IP assays were performed as previously described [13,15,16,23,24,25,26,27,28,29,30,31]. Briefly, mock vector as control vector, MYC-TRAF6, FLAG-AMPKαl, and MYC-BECN1 were transfected into HEK293T cells using Lipofectamine 2000 for 38–48 h. Cell lysates were immunoprecipitated with anti-FLAG or anti-MYC antibody. For interaction domain mapping, HEK293T cells were transfected with mock vector, MYC-AMPKαl, MYC-BECN1, MYC-TRAF6, FLAG-AMPKαl along with FLAG-TRAF6 wild type (WT) and FLAG-TRAF6 truncated mutants, FLAG-AMPKαl WT and FLAG-AMPKαl truncated mutants, or MYC-BECN1 WT and MYC-BECN1 truncated mutants using Lipofectamine 2000. At 38 h after transfection, cells were harvested, and cell lysates were immunoprecipitated with anti-MYC or anti-FLAG antibody. IP complexes were separated by SDS-PAGE (6–10%) and immune-probed with antibodies specific for anti-MYC, anti-FLAG, anti-TRAF6, or anti-AMPKα1. For ubiquitination assay, mock vector, MYC-BECN1, FLAG-TRAF6, or FLAG-AMPKα1 was transfected into HEK293T cells along with HA-tagged Ub. Cell lysates were immunoprecipitated with anti-MYC antibody and probed with antibodies specific for anti-MYC, anti-HA, anti-TRAF6, or anti-AMPKα1 antibody. Control (Ctrl) HEK293T and *AMPKα1*KO HEK293T cells were cultured in serum-free medium at different time periods. Cell lysates were immunoblotted with anti-LC3A/B antibody and anti-GAPDH as a loading control. Ctrl A549, *AMPKα1*KO A549, Ctrl MDA-MB-231, and *AMPKα1*KO MDA-MB-231 cells were treated with or without vehicle or CQ (10 μM) in the presence or absence of LPS (10 μg/mL) for 6 h. Whole cell lysates were immunoblotted with anti-LC3A/B antibody and anti-GAPDH as a loading control. Ctrl THP-1 and AMPKα1KD THP-1 cells were treated with or without LPS for 2 h. Whole cell lysates were immunoblotted with anti-LC3A/B, anti-TAK1, anti-pho-TAK1, and anti-GAPDH as a loading control.

### 2.9. Wound-Healing and Transwell Migration Assay

A wound-healing assay was performed as previously described [13,14,15,16]. Ctrl A549, *AMPKα1*KO A549, Ctrl MDA-MB-231, *AMPKα1*KO MDA-MB-231, Ctrl MCF-7, and *AMPKα1*KO MCF-7 cells were cultured in 12-well plates to reach confluence. Cell monolayers were gently scratched using a sterile yellow Gilson-pipette tip to make a wide gap (approximately 400 μm). Cells were washed with culture medium. Floating cells and debris were removed from plates. Cells were treated with vehicle (DMSO), 3-MA (5mM), CQ (10 μM) in the presence or absence of LPS (10 μg/mL). Images were captured after culturing for different time periods as indicated in each experiment. Transwell migration assay was performed as previously described [21,22]. Transwell inserts (8 μm pore; Corning, 3422) were sited into wells for cell migration assay. Ctrl A549, *AMPKα1*KO A549, Ctrl MDA-MB-231, *AMPKα1*KO MDA-MB-231, Ctrl MCF-7, and *AMPKα1*KO MCF-7 cells (5 × 10^4^ cells per well) were suspended in culture medium (RPMI) including vehicle, 3-MA (5mM), or CQ (10 μM) in the presence or absence of LPS (10 μg/mL) and placed into the top chambers of 24-transwell plates. Culture medium RPMI containing 10% FBS was added to bottom chambers. After an overnight incubation, non-migrated cells remaining in the top chamber were removed. Migrated cells found in the bottom chamber were fixed. To visualize nuclei, cells were stained using crystal violet. All experiments were performed in triplicate. Experiments were repeated twice.

### 2.10. Immunofluorescence Microscopy

Immunofluorescence microscopy to assess LC3 puncta was performed as previously described [15]. Briefly, Ctrl HEK293T and *AMPKα1*KO HEK293T cells were grown on glass coverslips overnight, fixed with 4% paraformaldehyde (Sigma, P-6148), and treated with 0.2% Triton X-100 (Sigma, T9284) to permeabilize for 30 min on ice. Cells to be blocked with 2% BSA for 1 h were treated with primary anti-LC3 antibody for 2 h, and further incubated with Alexa Fluor 488-conjugated donkey anti-rabbit IgG (Molecular Probes, A21206, 1:500 dilution) for 1 h at room temperature in the dark, washed with PBS, and stained with 1 μg/mL DAPI (Sigma, D9564). After being mounted with vecta shield mounting medium (Vector Laboratories, H-1000, Burlingame, CA, USA), slides were examined under a LSM 710 laser-scanning confocal microscope (Carl Zeiss, Jena, Germany).

### 2.11. GEPIA Data Analysis

Expression datasets of correlation between AMPKαl (*PRKAA1*) and *TRAF6*, TAK1 (*MAP3K7*), or *TLR4* in breast invasive carcinoma (BRCA tumor) were obtained from Gene Expression Profiling Interactive Analysis (GEPIA), a web-based tool (http://gepia.cancer-pku.cn/).

### 2.12. Statistical Analysis

In vitro data are expressed as mean ±SEM of triplicate samples. Statistical significance was analyzed using ANOVA or Student’s t-test of GraphPad Prism 5.0 (GraphPad Software, San Diego, CA, USA).

## 3. Results

### 3.1. AMPKα1 is Functionally Associated With TRAF6 and BECN1

Beclin1 (*BECN1*) is a key activator of early autophagosome assembly for autophagy induction through its ubiquitination by TRAF6 and phosphorylation by AMPK [9,10,11,12]. We first examined whether AMPKα1 was functionally associated with TRAF6-BECN1 complex. TRAF6 was co-precipitated with AMPKαl (Figure 1A, lane 4 in IP:MYC). To elucidate the molecular interaction between TRAF6 and AMPKαl, truncated mutants of TRAF6 and AMPKα1 were generated (Appendix A, TRAF6 and Appendix A, AMPKα1) and immunoprecipitation (IP) assay was performed with AMPKα1 (Figure 1B) or TRAF6 (Figure 1C). FLAG-TRAF6 wild type (WT), FLAG-tagged TRAF6 110-522, and FLAG-tagged TRAF6 260-522 were co-precipitated with MYC-AMPKα1 (Figure 1B, lanes 2–4), whereas no significant interaction could be seen for FLAG-TRAF6 349-522 (Figure 1B, lane 5). In addition, FLAG-AMPKα1 WT, FLAG- AMPKα1 1-392, and FLAG-AMPKα1 1-312 were co-precipitated with MYC-TRAF6 (Figure 1C, lanes 2–4), whereas no significant interaction could be seen for FLAG-AMPKα1 393-550 (Figure 1C, lane 5), indicating that the coiled-coil domain of TRAF6 could interact with the catalytic domain of AMPKα1, as depicted in Figure 1D. We next examined molecular interactions between AMPKα1 and BECN1. BECN1 was co-precipitated with AMPKα1 (Figure 1E, lane 4). MYC-BECN1 was significantly co-precipitated with truncated mutants of AMPKαl, along with wild type (WT) AMPKαl (Figure 1F, lanes 2–5). To elucidate the interaction site of BECN1 to AMPKαl, truncated mutants of BECN1 were generated (Appendix A) and IP assay was performed with AMPKα1. MYC-BECN1 WT and MYC-BECN1 1-269 were significantly co-precipitated with FLAG-AMPKα1 (Figure 1G, lanes 6 and 7), whereas no significant interaction could be seen for MYC-BECN1 1-127 (Figure 1G, lane 8), indicating that the coiled-coil domain of BECN1 could interact with the βγ-binding domain of AMPKα1, as depicted in Figure 1H.

### 3.2. AMPKα1 Enhances Ubiquitination of BECN1 and Regulates Autophagy Induction in Serum Starvation Condition

Based on the above results, we could propose molecular associations among TRAF6, AMPKα1, and BECN1 as depicted in Figure 2A. To confirm these molecular associations, IP assay was performed. As expected, MYC-BECN1 was co-precipitated with FLAG-TRAF6 in the absence of FLAG-AMPKα1 (Figure 2B, lane 2) and significantly co-precipitated with FLAG-TRAF6 in the presence of FLAG-AMPKα1 (Figure 2B, lane 3). Interestingly, the interaction between BECN1 and TRAF6 was enhanced in the presence of AMPKα1 compared to that in the absence of AMPKα1 (Figure 2B, lane 3 versus lane 2). These results suggest that AMPKα1 could facilitate the association of TRAF6 and BECN1. Through the endogenous immunoprecipitation assay, moreover, the molecular complex, TRAF6-BECN1-AMPKα1, could be observed in A549 cells in response to LPS stimulation (Appendix A, lane 4). Next, we asked whether AMPKα1 could affect ubiquitination of BECN1 by TRAF6. Consistently, ubiquitination of BECN1 was induced by TRAF6 in the absence of AMPKα1 (Figure 2C, lane 4). Interestingly, BECN1 ubiquitination was markedly enhanced in the presence of AMPKα1 (Figure 2C, lane 4 versus lane 5), suggesting that AMPKα1 could induce the stabilization of TRAF6-BECN1 complex and enhance the ubiquitination of BECN1 as depicted in Figure 2A. Recent evidence has revealed a pivotal role of post-translational modification of BECN1 in early autophagosome assembly through ubiquitination by TRAF6 and phosphorylation by AMPK [9,10,11,12], as indicated in Figure 2A. Furthermore, a previous report has shown that AMPK regulates autophagy by phosphorylating BECN1 at threonine 388 during glucose deprivation [11]. Therefore, we asked whether AMPKα1 might be functionally implicated in autophagy induction under a serum starvation condition. We generated *AMPKα1*-knockout (*AMPKα1*KO) HEK293T cells using CRISPR-Cas9 system (Appendix A). Upon serum starvation, levels of LC3-II were significantly attenuated in *AMPKα1*KO HEK293T cells than in Ctrl HEK293T cells (Figure 2D, Ctrl versus *AMPKα1*KO in LC3-II). Additionally, LC3 puncta were markedly decreased in *AMPKα1*KO HEK293T cells (Figure 2D, Ctrl vs. *AMPKα1*KO), suggesting a pivotal role of AMPKα1 in autophagy induction.

### 3.3. AMPKα1-Knockdown THP-1 Cells Show Impaired Autophagy Flux by TLR4 Stimulation

Recent evidence has shown that TLR3/4 stimulations can induce autophagy through the TRAF6-BECN1 signaling axis [14,15,16]. Moreover, a previous report has shown that AMPKα1 can regulate TLR4-mediated signaling through activation with TAK1 [8]. Since results described above showed that AMPKα1 could positively regulate the TRAF6-BECN1 axis for autophagy, we next asked whether AMPKα1 was functionally associated with autophagy flux including the initiation of autophagy and autophagosome biosynthesis by TLR4. To address this issue, we generated *AMPKα1*-knockdown (*AMPKα1*KD) human monocytic THP-1 cells using lentiviral particles containing shRNA for *AMPKα1* gene (Figure 3A) as described in Materials and Methods. Upon TLR4 stimulation, the activation of TAK1 and the level of LC3-II were attenuated in *AMPKα1*KD THP-1 cells than in Ctrl THP-1 cells (Figure 3B, lane 2 versus lane 4). Furthermore, AMPKα1 WT overexpression induced an increase of LC3-II level, whereas AMPKα1 (D159A) DN (dominant negative) showed no significant change in LC3-II level with or without LPS treatment (Figure 3C, DN versus WT), supporting that the kinase activity of AMPKαl was critical for TLR4-induced autophagy induction. Based on these results, Ctrl THP-1 and *AMPKα1*KD THP-1 cells were treated with or without LPS and microarray analysis was performed. As shown in Figure 3D, 13 ATG-related genes for autophagy formation were significantly up-regulated in Ctrl THP-1 cells treated with LPS (Figure 3D, red bars in column 1), whereas eight genes were down-regulated in *AMPKα1*KD THP-1 cells treated with LPS (Figure 3D, green bars in column 2). When gene expression levels were compared between *AMPKα1*KD THP-1 cells treated with LPS and Ctrl THP-1 cells treated with LPS, six genes were significantly down-regulated in *AMPKα1*KD THP-1 cells (Figure 3D, green bars in column 3). Several functional units, such as ATG9A system, ATG12-ATG5 conjugation system, and ATG12-conjugation system, are critically involved in assembly and formation of autophagosome [32]. In addition, the ATG12, activating by the E1 enzyme ATG7, is conjugated to ATG5 via the E2 enzyme ATG10 and then the ATG12-ATG5 conjugate can be stabilized by ATG16L proteins and further form ATG12-ATG5-ATG16L complex [32,33,34,35]. Based on these previous reports, we supposed that *AMPKα1* might be implicated in the expression of ATG7, ATG9A, and ATG12, thereby involved in the autophagy formation. Additionally, 21 genes related to autophagosome biogenesis were up-regulated in Ctrl THP-1 cells treated with LPS and 15 genes were down-regulated in *AMPKα1* KD THP-1 cells treated with LPS. Moreover, 13 genes were down-regulated in *AMPKα1*KD THP-1 compared to those in Ctrl THP-1 in the presence of LPS (Figure 3E, red bars in column 1 and green bars in column 2 and column 3, respectively). Phosphatidylinositol 3-phosphate (PtdIns3P) produced by PI3K complex I is detected in the isolation membrane and autophagosomal membrane, thereby involved in the autophagosomal biosynthesis process [36,37]. PtdIns3P binds to its effector proteins including ZFYVE1 or WIPI (WD-repeat protein interacting with phosphoinositides), and that functions as a platform for the formation of the phagophore [36,37]. To see whether AMPKα1 is implicated in the autophagosomal biosynthesis process, genes to be related to the process were further assessed. Twenty-one genes related to autophagosome biogenesis were up-regulated in Ctrl THP-1 cells treated with LPS and 15 genes were down-regulated in *AMPKα1*KD THP-1 cells treated with LPS. Moreover, 13 genes were down-regulated in *AMPKα1*KD THP-1 compared to those in Ctrl THP-1 in the presence of LPS (Figure 3E, red bars in column 1 and green bars in column 2 and column 3, respectively), indicating that *AMPKα1* affects the PtdIns3P-mediated pathway for autophagosome biogenesis. These results suggest that AMPKαl is associated with the autophagy flux for autophagy formation induced by TLR4 stimulation.

### 3.4. AMPKα1 is Positively Associated With Autophagy in AMPKα1-Knockout A549 Human Lung Cancer and Primary-NSCLCs

Much evidence demonstrates that autophagy can promote cancer progression in many different cancers [17,18,19,20]. In the present study, we assessed whether AMPKα1 could affect cancer progression by regulating autophagy. First, *AMPKα1*KO A549 human lung cancer cells were generated (Figure 4A). Upon TLR4 stimulation, LC3-II level was increased in Ctrl A549 cells but significantly attenuated in *AMPKα1*KO A549 cells (Figure 4B,C, Ctrl A549 versus *AMPKα1*KO A549 treated with LPS). As expected, treatment with chloroquine (CQ), an autophagy inhibitor that blocks the binding of autophagosomes to lysosomes, induced increases of LC3-II in both cells (Figure 4B,C, LPS plus CQ in Ctrl A549 and *AMPKα1*KO A549). To assess the role of AMPKα1 in cancer cell migration and invasion, Ctrl A549 and *AMPKα1*KO A549 cells were treated with or without vehicle, LPS, or CQ, as indicated in Figure 4D. Upon LPS stimulation, the invasion ability of *AMPKα1*KO A549 cells was decreased in response to LPS than that of Ctrl A549 cells (Figure 4D,E, Ctrl A549 versus *AMPKα1*KO A549 cells treated with LPS). TLR4-induced autophagy activation promoted migration and invasion of lung cancer by induction of chemokines and immunosuppressive factors, such as IL-6, MMP2, and CCL2 [14]. Interestingly, the levels of IL-6 mRNA, MMP2 mRNA, and CCL2 mRNA were significantly decreased in *AMPKα1*KO A549 cells, as compared with those of Ctrl A549 cells (Appendix A, *AMPKα1*KO A549 treated with LPS versus Ctrl A549 treated with LPS). As expected, the treatment of an autophagy inhibitor 3-MA in the presence of LPS attenuated the levels of IL-6 mRNA, MMP2 mRNA, and CCL2 mRNA (Appendix A, LPS versus LPS + 3-MA). Similar results were observed for the migration capacity assessed by wound healing assay (Figure 4F,G, Ctrl A549 versus *AMPKα1*KO A549 cells treated with LPS), suggesting that AMPKα1 was functionally involved in cancer cell migration and invasion.

Having obtained results shown above, we tried to find clinical evidence using tumor and matched normal tissues from 42 NSCLC patients [21], as described in Appendix A. We analyzed gene expression using Human HT-12 expression BeadChips as described in Materials and Methods. Five of forty-two tumor tissues showed significantly up-regulated expression of *AMPKαl* (*PRKAA1*) (Group 1) (Figure 5A, red bars), whereas another six tumor tissues showed significantly down-regulated expression of *AMPKαl* (*PRKAA1*) (Group 2) (Figure 5A, green bars). We then compared gene expression levels between these two groups. Interestingly, the levels of autophagy-related genes were positively associated with group 1 tumor tissues (Figure 5B, *PRKAA1* up-regulated patients), but negatively associated with group 2 tumor tissues (Figure 5B, *PRKAA1* down-regulated patients). Furthermore, cell migration-, adhesion-, and metastasis-related genes were mostly up-regulated in group 1, but markedly decreased in group 2 (Figure 5C, group 1 versus group 2). Since AMPKα1was functionally associated with TLR4-mediated signaling through activation of TAK1 [8], we further analyzed expression levels of TLRs-related genes such as TLRs, cytokine-related genes, *NF-κB* genes, *IRF* genes, and *NF-κB*-dependent genes. Importantly, levels of *TLR1, TLR10, TLR3, TLR4, TLR5, TLR6, TLR7*, and *TLR8* were significantly up-regulated in group 1 cancer tissues expected for patient 7, but markedly down-regulated in group 2 tumor tissues (Figure 5D, group 1 versus group 2). Similar results were observed for cytokine-related genes, *NF-κB* genes, *IRF* genes, and *NF-κB*-dependent genes (Figure 5E, cytokines-related genes; Figure 5F, *NF-κB* genes; Figure 5G, *IRF* genes; Figure 5H, *NF-κB*-dependent genes: group 1 versus group 2). These results strongly support that AMPKαl is positively associated with the regulation of autophagy and TLRs-mediated signaling.

### 3.5. AMPKα1-Knockout MDA-MB-231 and MCF-7 Breast Cancer Cells Show Attenuated Cancer Migration and Invasion

Based on results shown above, we tried to further verify the function of AMPKαl in another cancer. We generated *AMPKα1*KO MDA-MB-231 and *AMPKα1*KO MCF-7 human breast cancer cells (Appendix A, *AMPKα1*KO MDA-MB-231; Appendix A, *AMPKα1*KO MCF-7). Similar to results obtained for *AMPKα1*KO A549 cells, levels of LC3-II were also decreased in *AMPKα1*KO MDA-MB-231 and *AMPKα1*KO MCF-7 cells treated with LPS (Appendix A, Ctrl MDA-MB-231 or Ctrl MCF7 versus *AMPKα1*KO MDA-MB-231 or *AMPKα1*KO MCF-7 cells, respectively). Additionally, the migration and invasion capacity of *AMPKα1*KO MDA-MB-231 cells were consistently attenuated than those of Ctrl MDA-MB-231 cells (Figure 6A–C, Ctrl MDA-MB-231 versus *AMPKα1*KO MDA-MB-231). Consistent with these results, the migration and invasion capacity of *AMPKα1*KO MCF-7 cells were attenuated than those of MCF-7 cells (Figure 6D–F, Ctrl MCF-7 versus *AMPKα1*KO MCF-7), supporting the functional role of AMPKα1 in cancer migration and invasion. AMPKα1 is involved in TLR4-mediated signaling through the regulation of TAK activity [8]. Thus, we analyzed whether AMPKα1 was correlated with the expression of TRAF6, TAK1, and TLR4. Correlation analysis provided by gene expression profiling interactive analysis (GEPIA) revealed that the expression of AMPKα1 was significantly correlated with the expression of TRAF6, TAK1, and TLR4 in human breast cancers (Figure 7A, TRAF6; Figure 7B, TAK1; Figure 7C, TLR4). Similar correlations were also observed in human lung cancers (Appendix A). Together, these results suggest that AMPKα1 can regulate cancer cell migration and invasion by regulating autophagy induction.

## 4. Discussion

AMPK is a member of serine/threonine kinase, and forms a catalytic subunit (α1 or α2) and two regulatory heterotrimeric as subunit β (β1 or β2) and γ (γ1, γ2 or γ3) [1]. Each AMPK complex is composed by one α-subunit, one β-subunit, and γ regulatory subunit with critical roles in various cellular responses, such as autophagy [2,3], inflammation [4,5], and immunity [6,7,8]. AMPKα1 as an activating kinase of TAK1 has a key role in mediating inflammatory signals triggered by TLR4 [8]. However, the mechanism of AMPKαl on cancer progression through TLR4-mediated autophagy activation has not yet been investigated. In the current study, we report that AMPKαl is functionally involved in autophagy induction. It also regulates cancer cell migration and invasion induced by TLR4 stimulation. Through biochemical studies, we found that AMPKαl interacted with TRAF6 and BECN1 proteins and induced the enhancement of BECN1 ubiquitination, suggesting that AMPKαl could regulate the TRAF6-BECN1 signaling axis for autophagy induction. Additionally, we found that *AMPKα1*KO cancer cells such as *AMPKα1*KO A549, *AMPKα1*KO MDA-MB-231, and *AMPKα1*KO MCF-7 cells showed significantly attenuated TLR4-induced migration and invasion. Importantly, microarray analysis of *AMPKα1*KD THP-1 cells and patient-derived NSCLCs revealed that AMPKαl expression was positively associated with the expression of autophagy flux-related genes. These results suggest that autophagy regulation by AMPKαl was functionally associated with cancer progression.

Post-translational modifications such as phosphorylation and ubiquitination of BECN1 are critical for the autophagy flux and induction in response to various cellular stimuli [9,10,11,12]. Upon TLR4 stimulation, TRAF6 can interact with BECN1 and induce TRAF6-mediated lysine (K) 63-linked ubiquitination of BECN1, thereby controlling autophagy formation and inflammatory responses [13,14,15,16]. In addition, AMPK induces phosphorylation of BECN1 at different sites such as its S90, S93, and T388 to regulate autophagy [11]. Under the condition of glucose deprivation, cellular AMPK is activated to induce phosphorylation BECN1 at T388, leading to the induction of autophagy [11]. This suggests that AMPK is implicated in the regulation of autophagy through phosphorylation of BECN1. The phosphorylation of BECN1 at threonine 388 induced by AMPK is critical for autophagy regulation [11]. We found that AMPKαl enhanced the ubiquitination of BECN1 through the formation of molecular complex TRAF6-AMPKαl-BECN1. Moreover, a mutant of AMPKαl, AMPKαl (D159A) DN, resulted in the attenuation of LC3-II level induced by TLR4. Although we do not have a direct evidence for the phosphorylation of T388 BECN1 by AMPKαl because the antibody is not commercially available, AMPKαl might induce the phosphorylation of BECN1 and regulate the autophagy formation accompanied by an enhancement of BECN1 ubiquitination based on results of the present study and a previous report [11].

Accumulating evidence has shown that autophagy is functionally linked to cancer progression such as migration, invasion, and metastasis [17,18,19,20]. Recently, it has been shown that engagement of TLR ligands can induce cancer migration and invasion by regulating the TRAF6-BECN1 signaling axis [13,14,15,16]. Moreover, it has been reported that AMPKαl regulates TLR4-mediated signaling through the activation of TAK1 [8]. Based on these results, we speculated that AMPKαl could modulate cancer cell progression by regulating the TRAF6-BECN1 signaling axis. Therefore, we tested the capacity of cancer cell migration and invasion in *AMPKα1*-deficient cancer cells. In line with previous results [13,14,15,16], our results revealed that the migration and invasion of *AMPKαl*KO A459 lung cancer, *AMPKα1*KO MDA-MB-231 breast cancer, and *AMPKα1*KO MCF-7 breast cancer cells were significantly attenuated by TLR4 stimulation. To further verify the regulatory role of AMPKαl in autophagy induction in cancer cells, we performed microarray analysis for *AMPKα1*KD THP-1 and patient-derived NSCLCs. Interestingly, TLR4-induced autophagy-related genes and autophagosome biosynthesis-related genes were down-regulated in *AMPKα1*KD THP-1 cells. In addition, expression levels of these genes were significantly decreased in patient-derived NSCLCs with low expression of AMPKαl than in patient-derived NSCLCs with high expression of AMPKαl. These results suggest that AMPKαl might be functionally associated with cancer progression, presumably through the regulatory mechanism of AMPKαl in the TRAF6-BECN1 signaling axis for autophagy induction presented in this study. According to previous reports [38,39], nevertheless, there are some controversial aspects regarding the positive or negative effect of AMPK activity on cancer migration. The phosphorylation of PDZ (Postsynaptic density 95, discs large and zonula occludens-1) and LIM Domain 5 (Pdlim5) on Ser177 by AMPK regulates inhibition of cell migration by suppressing the Rac1-Arp2/3 signaling pathway [38], suggesting that increased AMPK activity is negatively implicated in cell migration. In contrast, it has been reported that activation of AMPK by Lysophosphatidic acid (LPA) induced cell migration through the signaling pathway to cytoskeletal dynamics and increases tumor metastasis in ovarian cancer [39]. At least two possibilities are likely to suggest the specificity according to AMPK activity regulated by extracellular stimulations and the specific regulation by AMPKα1. In order to clarify this, more diverse and detail mechanism studies are required in the future.

## 5. Conclusions

In conclusion, this study proposes a mechanistic model for how AMPKαl regulates autophagy induction, thereby causing cancer progression as depicted in Figure 7F. AMPKαl interacts with TRAF6 and BECN1 to form a tri-molecular complex, TRAF6-AMPKαl-BECN1 (Figure 7F, left). This molecular association induces post-translational modification of BECN1 such as ubiquitination by TRAF6 and phosphorylation by AMPKαl (Figure 7F, middle). Such post-translational modification of BECN1 eventually leads to the activation of autophagy flux through the PI3K-II and ULK complex and autophagy (Figure 7F, right). Finally, this study provides evidence for the functional involvement of AMPKαl in NSCLC, thereby proposing a possible model of how AMPKαl is implicated in cancer progression through the regulation of autophagy.

## Figures and Tables

**Figure 1 cancers-12-03289-f001:**
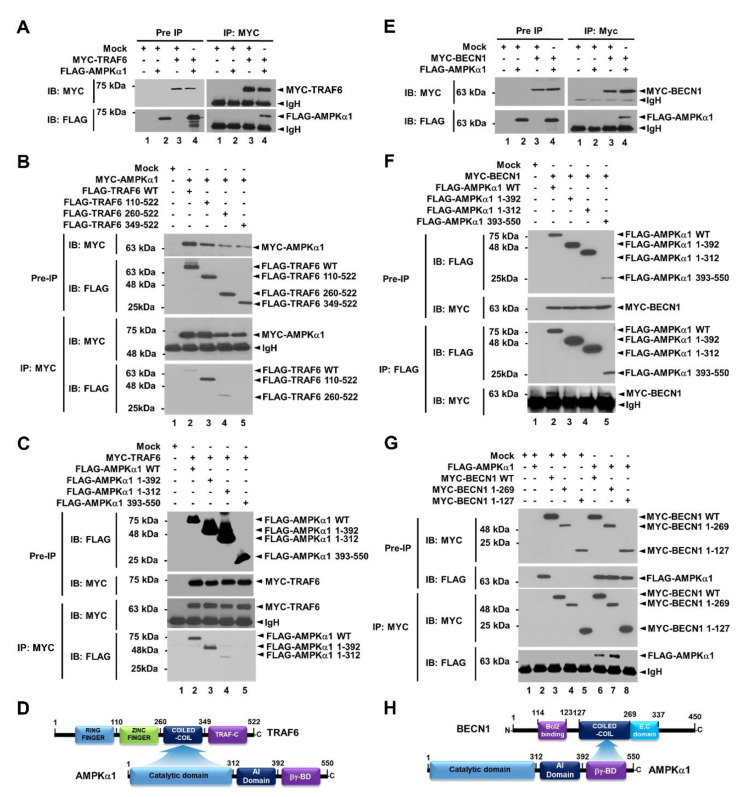
AMPKαl interacts with TRAF6 and BECN1 proteins. (**A**) Mock control vector, MYC-TRAF6, or FLAG-AMPKαl was transfected into HEK293T cells as indicated. Immunoprecipitation (IP) assay was performed with anti-MYC antibody. Immuno-blot membrane was probed with antibodies specific for MYC or FLAG. (**B**) HEK293T cells were transfected with mock vector, MYC-AMPKαl, FLAG-TRAF6 wild type (WT), or FLAG- truncated mutants of TRAF6 as indicated. IP assay was performed with anti-MYC antibody. IP samples were separated by SDS-PAGE and probed with anti-MYC or anti-FLAG antibody. (**C**) HEK293T cells were transfected with mock vector, MYC-TRAF6, FLAG-AMPKαl WT, or FLAG-truncated mutants of AMPKαl as indicated. IP assay was performed with anti-MYC antibody. IP samples were separated by SDS-PAGE and probed with anti-MYC or anti-FLAG antibody. (**D**) A schematic view of molecular interactions between TRAF6 and AMPKαl. (**E**) Mock control vector, MYC-BECN1, or FLAG-AMPKαl was transfected into HEK293T cells as indicated. IP assay was performed with anti-MYC antibody. Immuno-blot membrane was probed with antibodies specific for MYC or FLAG. (**F**) HEK293T cells were transfected with mock vector, MYC-BECN1, FLAG-AMPKαl WT, or FLAG-truncated mutants of AMPKαl as indicated. IP assay was performed with anti-FLAG antibody. IP samples were separated by SDS-PAGE and probed with anti-MYC or anti-FLAG antibody. (**G**) HEK293T cells were transfected with mock vector, FLAG-AMPKαl, MYC-BECN1 WT, or MYC-truncated mutants of BECN1 as indicated. IP assay was performed with anti-MYC antibody. IP samples were separated by SDS-PAGE and probed with anti-MYC or anti-FLAG antibody. (**H**) A schematic view of molecular interactions between BECN1 and AMPKαl. Uncropped western blot images are available in Appendix A.

**Figure 2 cancers-12-03289-f002:**
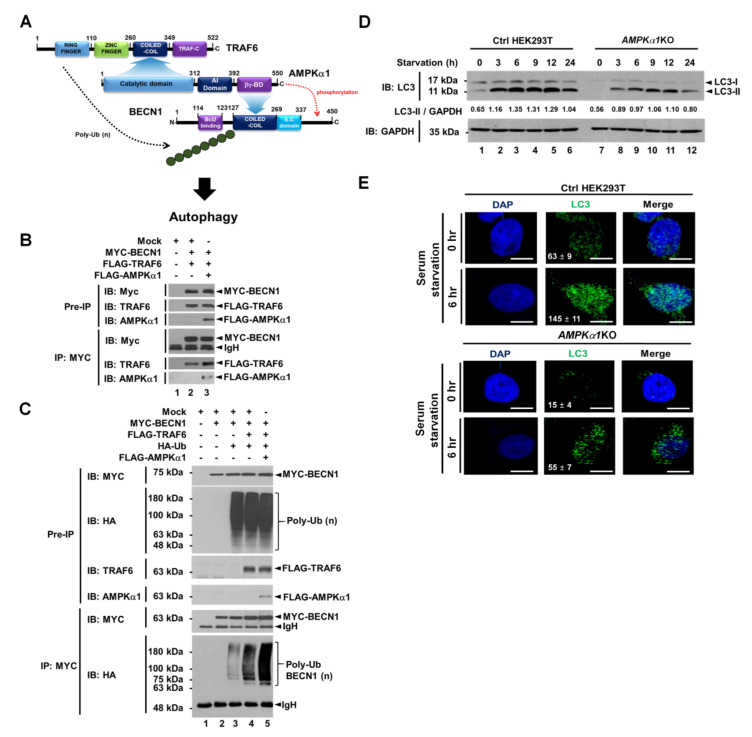
AMPKαl enhances ubiquitination of BECN1 and autophagy under serum starvation. (**A**) A schematic view of molecular interactions among TRAF6, AMPKαl, and BECN1, and how ubiquitination and phosphorylation of BECN1 are regulated by TRAF6 and AMPKαl, respectively, for autophagy induction. (**B**) Mock vector, MYC-BECN1, FLAG-TRAF6, and FLAG-AMPKαl were transfected into HEK293T cells as indicated. IP assay was performed with anti-MYC antibody. IP samples were separated by SDS-PAGE and probed with anti-MYC, anti-TRAF6, or anti-AMPKαl antibody. (**C**) Mock vector, MYC-BECN1, FLAG-TRAF6, FLAG-AMPKαl, and HA-Ub were transfected into HEK293T cells as indicated. IP assay was performed with anti-MYC antibody. IP samples were separated by SDS-PAGE and probed with anti-MYC, anti-HA, anti-TRAF6, or anti-AMPKαl antibody. (**D**) Ctrl and *AMPKα1*KO HEK293T cells were cultured in serum-free medium for different time periods as indicated. Cell lysates were immunoblotted with anti-LC3-I/-II antibody and anti-GAPDH antibody as a loading control. LC3-II levels were analyzed with Image J program. (**E**) Ctrl and *AMPKα1*KO HEK293T cells were cultured in serum-free medium for 6 h and then fixed. Immunofluorescence assay was performed with anti-LC3 antibody. Digital images were captured with confocal microscopy and the number of LC3-puncta was scored. Quantification represents the mean ±SEM of puncta per cell (*n* = 5) from 3 independent experiments. Scale bar: 10 μm. Uncropped western blot images are available in Appendix A.

**Figure 3 cancers-12-03289-f003:**
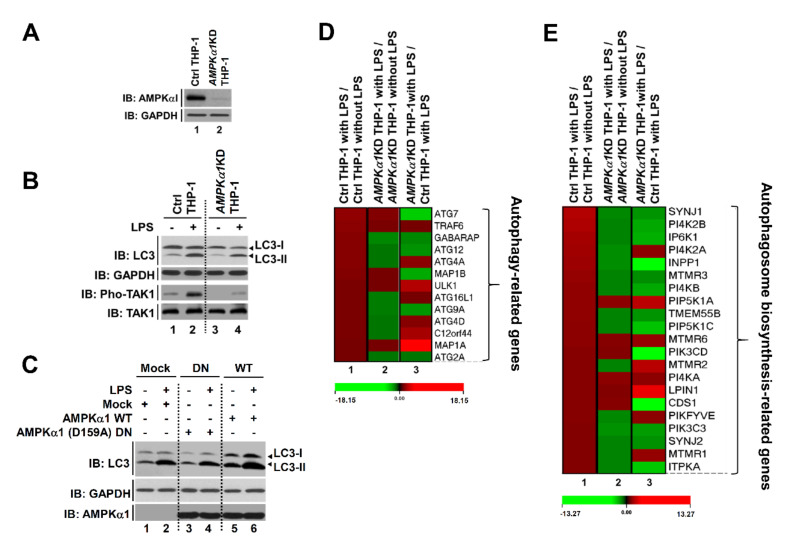
*AMPKα1*-knockdown THP-1 cells show impaired autophagy flux induced by TLR4 stimulation. (**A**) *AMPKα1*KD THP-1 cells were generated as described in Materials and Methods. Knockdown efficacy of AMPKαl was confirmed by western blotting with anti-AMPKαl antibody. (**B**) Control (Ctrl) and *AMPKα1*KD THP-1 cells were treated with or without LPS for 2 h. Cell lysates were immunoblotted with antibodies specific for LC3-I/-II, GAPDH, phosphorylated TAK1 (pho-TAK1), or TAK1. (**C**) HEK293T cells were transfected with mock, AMPKαl WT, or AMPKα1 (D159A) DN as indicated. Cells were treated with or without LPS. Cell lysates were immunoblotted with antibodies specific for LC3-I/-II, AMPKαl, or GAPDH. (**D**,**E**) Ctrl and *AMPKα1*KD THP-1 cells were treated with or without LPS for 6 h. Expression levels of autophagy-related Atg genes (13 different genes, (**D**)) and autophagosome synthesis-related genes (21 different genes, (**E**)) were compared between Ctrl THP-1 cells treated with LPS and Ctrl THP-1 cells without LPS treatment (lane 1), between *AMPKα1*KD THP-1 cells treated with LPS and *AMPKα1*KD THP-1 cells without LPS treatment (lane 2), or between *AMPKα1*KD THP-1 cells treated with LPS and Ctrl THP-1 cells treated with LPS (lane 3). ATG7, Autophagy Related 7; GABARAP, Gamma-aminobutyric acid receptor-associated protein; ATG12, Autophagy Related 12; ATG4A, Autophagy Related 4A Cysteine Peptidase; MAP1B, Microtubule-associated protein 1B; ULK1, Unc-51 Like Autophagy Activating Kinase 1; ATG16L1, Autophagy Related 16 Like 1; ATG9A, Autophagy-related protein 9A; ATG4D, Autophagy Related 4D Cysteine Peptidase; C12orf44, Autophagy-related protein 101; MAP1A, Microtubule Associated Protein 1A; ATG2A, Autophagy Related 2A; PI4K2B, Phosphatidylinositol 4-kinase type 2-beta; IP6K1, Inositol Hexakisphosphate Kinase 1; PI4K2A, Phosphatidylinositol 4-Kinase Type 2 Alpha; INPP1, Inositol Polyphosphate-1-Phosphatase; MTMR3, Myotubularin-related protein 3; PI4KB, Phosphatidylinositol 4-Kinase Beta; PIP5K1A, Phosphatidylinositol-4-Phosphate 5-Kinase Type 1 Alpha; PIP5K1C, Phosphatidylinositol-4-Phosphate 5-Kinase Type 1 Gamma; MTMR6, Myotubularin Related Protein 6; PIK3CD, Phosphatidylinositol-4,5-Bisphosphate 3-Kinase Catalytic Subunit Delta; PI4KA, Phosphatidylinositol 4-kinase alpha; CDS1, CDP-Diacylglycerol Synthase 1; PIKFYVE, Phosphoinositide Kinase, FYVE-Type Zinc Finger Containing; PIK3C3, Phosphatidylinositol 3-Kinase Catalytic Subunit Type 3; MTMR1, Myotubularin Related Protein 1; ITPKA, Inositol-Trisphosphate 3-Kinase A. Uncropped western blot images are available in Appendix A.

**Figure 4 cancers-12-03289-f004:**
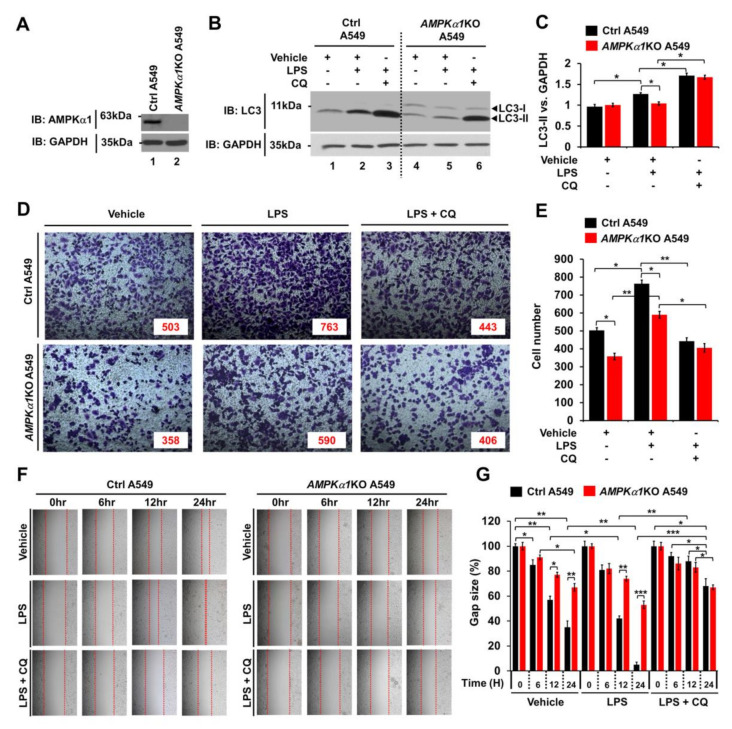
*AMPKα1*KO A549 cells show inhibition of autophagy induction and decreased cell migration in response to TLR4 stimulation. (**A**) *AMPKα1*KO A549 cells were generated as described in Materials and Methods. Knockout efficacy of AMPKαl was confirmed by western blotting with anti-AMPKαl antibody. (**B**,**C**) Control (Ctrl) A549 and *AMPKα1*KO A549 cells were treated with or without LPS in the presence or absence of CQ as indicated. Cell lysates were immunoblotted with antibodies specific for LC3-I/-II or GAPDH (**B**). LC3-II levels were analyzed with Image J program (**C**). Data shown are average values (±SEM) from a minimum of three independent experiments. **p* < 0.05. (**D**,**E**) Ctrl and *AMPKα1*KO A549 cells were suspended in RPMI medium including vehicle, LPS (10 μg/mL), or CQ (10 μM) plus LPS (10 μg/mL) and placed in top chambers of 24-transwell plates. After an overnight incubation, cells were fixed and stained with crystal violet (**D**). Numbers of migrating cells were counted. Results are presented as mean ±SEM of three independent experiments (**E**). **p* < 0.05. (**F**,**G**) Ctrl and *AMPKα1*KO A549 cells were seeded into 12-well cell culture plates. Confluent monolayers were scraped with a sterile yellow Gilson-pipette tip. The wound was then treated with vehicle (DMSO, <0.2% in culture medium), LPS (10 μg/mL), or CQ (10 μM) plus LPS (10 μg/mL) for different time periods as indicated. A representative experiment is shown (**F**). The residual gap between migrating cells from the opposing wound edge was expressed as a percentage of the initial scraped area (±SEM, *n* = 3) (**G**). * *p* < 0.05; ** *p* < 0.01, *** *p* < 0.001. Uncropped western blot images are available in Appendix A.

**Figure 5 cancers-12-03289-f005:**
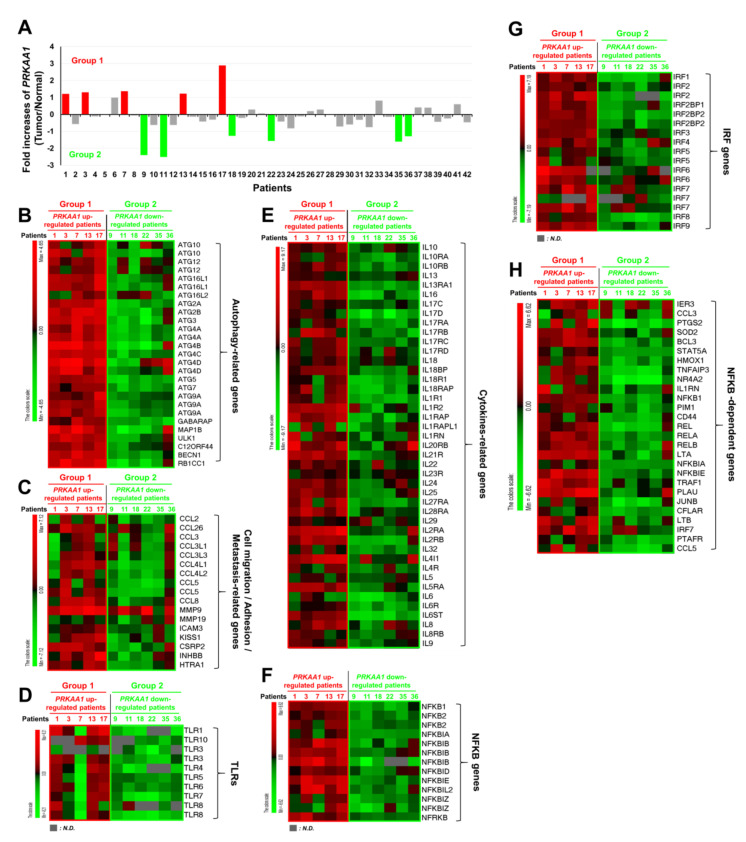
*AMPKα1* expression is positively associated with autophagy flux-related genes in primary non-small cell lung cancers. (**A**) Microarray analysis was performed for tumor and matched normal tissues from 42 NSCLC patients as described in Materials and Methods. Fold increases of *AMPKα1* (*PRKAA1*) expression in 42 NSCLs versus normal tissues are presented. According to increase or decrease of AMPKαl expression, tumors were divided into group 1 (red bars, *n* = 5) or group 2 (green bars, *n* = 6), respectively. (**B**–**H**) Bioinformatics analysis of microarray data between Group 1 and Group 2 was performed. Comparative analysis of the expression of autophagy-related genes (**B**), comparative analysis of the expression of cell migration/adhesion/metastasis-related genes (**C**), comparative analysis of the expression of TLR genes (**D**), comparative analysis of the expression of cytokines-related genes (**E**), comparative analysis of the expression of *NF-κB* genes (**F**), comparative analysis of the expression of IRF genes (**G**), comparative analysis of the expression of *NF-κB*-dependent genes (**H**) are shown. LogFC (fold change) values are presented. Red, up-regulated genes; Green, down-regulated genes.

**Figure 6 cancers-12-03289-f006:**
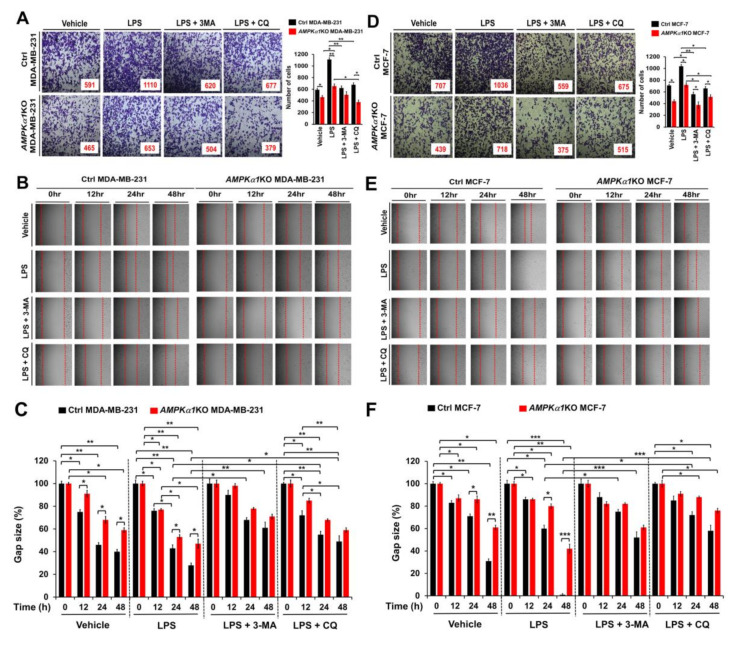
*AMPKα1*KO MDA-MB-231 and *AMPKα1*KO MCF-7 cells exhibit attenuated cell migration and invasion induced by TLR4. (**A**) Ctrl MDA-MB-231 and *AMPKα1*KO MDA-MB-231 cells were suspended in RPMI medium including vehicle, LPS (10 μg/mL), 3-MA (5 mM) plus LPS (10 μg/mL), or CQ (10 μM) plus LPS (10 μg/mL) and placed in top chambers of 24-transwell plates. After an overnight incubation, cells were fixed and stained with crystal violet (left). The numbers of migrating cells were counted. Results are presented as mean ±SEM of three independent experiments (right). (**B**,**C**) Ctrl MDA-MB-231 and *AMPKα1*KO MDA-MB-231 cells were seeded into 12-well cell culture plates. Confluent monolayers were scraped with a sterile yellow Gilson-pipette tip. The wound was then treated with vehicle (DMSO, <0.2% in culture medium), LPS (10 μg/mL), 3-MA (5 mM) plus LPS (10 μg/mL), or CQ (10 μM) plus LPS (10 μg/mL) for different time periods as indicated. A representative experiment is shown (**B**). The residual gap between migrating cells from the opposing wound edge was expressed as a percentage of the initial scraped area (±SEM, *n* = 3) (**C**). (**D**) Ctrl MCF-7 and *AMPKα1*KO MCF-7 cells were suspended in RPMI medium including vehicle, LPS, 3-MA plus LPS, or CQ plus LPS and placed in top chambers of 24-transwell plates. After an overnight incubation, cells were fixed and stained with crystal violet (left). The numbers of migrating cells were counted. Results are presented as mean ±SEM of three independent experiments (right). (**E**,**F**) Ctrl MCF-7 and *AMPKα1*KO MCF-7 cells were seeded into 12-well cell culture plates. Confluent monolayers were scraped with a sterile yellow Gilson-pipette tip. The wound was then treated with vehicle (DMSO, <0.2% in culture medium), LPS, 3-MA plus LPS, or CQ plus LPS for different time periods as indicated. A representative experiment is shown (**E**). The residual gap between migrating cells from the opposing wound edge was expressed as a percentage of the initial scraped area (±SEM, *n* = 3) (**F**). * *p* < 0.05; ** *p* < 0.01; *** *p* < 0.001.

**Figure 7 cancers-12-03289-f007:**
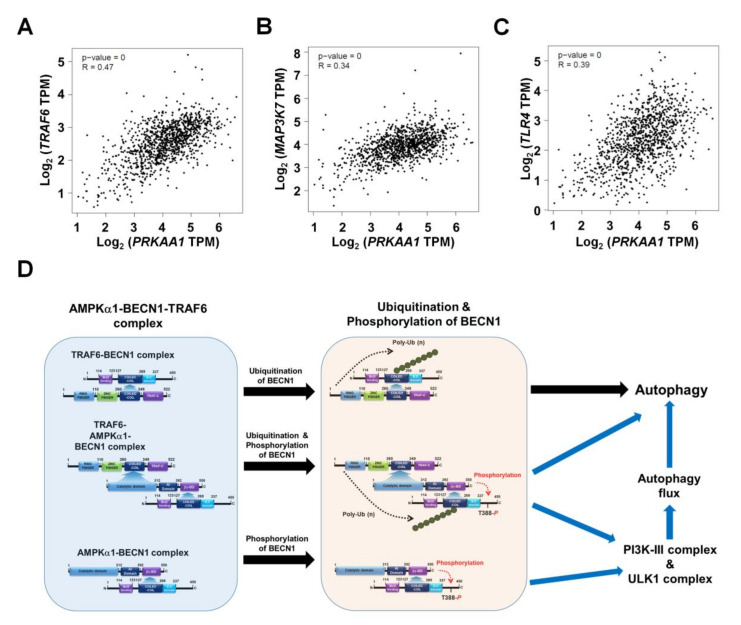
Correlation between AMPKαl (*PRKAA1*) and TRAF6, TAK1 (*MAP3K7*), or TLR4 in breast cancers. (**A**–**C**) Correlation between AMPKαl (*PRKAA1*) and TRAF6 (**A**), TAK1 (*MAP3K7*) (**B**), or TLR4 (**C**) in breast cancers revealed by GEPIA. (**D**) Schematic representation showing the proposed mechanism involved in the regulation of autophagy flux by AMPKαl for the induction of autophagy.

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
