# Peer review of "AMPKα1 Regulates Lung and Breast Cancer Progression by Regulating TLR4-Mediated TRAF6-BECN1 Signaling Axis"

_cancers, 2020, doi:10.3390/cancers12113289_

Round 1

Reviewer 1 Report

The manuscript by Kim et al. demonstrates that AMPKα1 is positively involved in autophagy induction and cancer progression by regulating the ubiquitination and phosphorylation of BECN1 through association with TRAF6-BECN1 complex. Specific comments are as follows:

  1. In the title, “AMPKα1 regulates cancer progression” are suggested to be replaced with “AMPKα1 regulates lung and breast cancer progression”.

  1. On page 8 and 19, “post-transcriptional modification of BECN1” should be changed to “post-translational modification of BECN1”.

  1. In Figure 2, Fig. 2A and 2D are suggested to be merged into one panel. In addition, AMPK was shown to phosphorylate Beclin 1 at S93, S96 (Kim et al., Cell 2013, 152:290-303), and T388 (Zhang et al., Autophagy 2016, 12:1447-1459) in response to glucose starvation. In order to demonstrate that AMPKα1 phosphorylates Beclin 1 at T388 in Fig. 2D in the current study, the authors should show the T388 phosphorylation levels of Beclin 1 in Ctrl and AMPKα1KO HEK293T cells upon serum starvation in Fig. 2E, and explain the reasons for using serum starvation rather than glucose starvation in these experiments.

  1. In Figure 3, the results for Fig. 3C should be rephrased as “Furthermore, AMPKα1 WT overexpression induced an increase of LC3-II level, whereas AMPKα1 (D159A) DN (dominant negative) showed no significant change in LC3-II level with or without LPS treatment (Figure 3C, DN versus WT)”. The results for Fig. 3E should be rephrased as “Additionally, 21 genes related to autophagosome biogenesis were up-regulated in Ctrl THP-1 cells treated with LPS and 15 genes were down-regulated in AMPKα1 KD THP-1 cells treated with LPS”. Please also explain more clearly about the results and implications of Fig. 3D and 3E. For example, please explain why eight ATG-related genes were “down-regulated” in AMPKα1 KD THP-1 cells treated with LPS (Fig. 3D, column 2)? When gene expression levels were compared between AMPKα1 KD THP-1 cells treated with LPS and Ctrl THP-1 cells treated with LPS, why did these six genes (ATG7, GABARAP, ATG12, MAP1B, ATG9A, and ATG2A) down-regulated in AMPKα1 KD THP-1 cells treated with LPS (Fig. 3D, column 3) differ from those eight genes (GABARAP, ATG12, ATG4A, ATG16L1, ATG9A, ATG4D, C12orf44, and ATG2A) down-regulated in AMPKα1 KD THP-1 cells treated with LPS (Fig. 3D, column 2)? There were similar questions for Fig. 3E. Please spell the terms “ATG7, GABARAP, MAP1B, ATG12, ATG4A, ATG16L1, ATG9A, ATG4D, C12orf44, ATG2A, etc.” in full in the figure legend.

  1. In Figure 4, please add some descriptions about the implications by using CQ to inhibit autophagy and induce increases of LC3-II in both Ctrl A549 and AMPKα1 KO A549 cells.

  1. In Figure 7, why could “the expression of AMPKα1 was markedly increased in lung metastatic breast cancer than in lymph node metastatic breast cancer (Figure 7B, Group A versus Group B)” suggest that “the expression of AMPKα1 might be positively associated with metastasis of breast cancers”? Both lung metastatic breast cancer and lymph node metastatic breast cancer are metastatic breast cancers.

  1. In the discussion, please discuss why only study the role of AMPKα1 in the regulation of autophagy, cancer migration, and invasion induced by TLR4 stimulation rather than AMPKα2.

  1. In the discussion, since augmented AMPK activity is shown to inhibit cell migration and prevent cancer metastasis (Yan et al., Nature Communications 2015, 6: 6137), while another study shows that activation of AMPK is essential for lysophosphatidic acid-induced cell migration in ovarian cancer cells (Kim et al., JBiol Chem 2011, 286(27):24036-45), please discuss the controversial role of AMPK in the regulation of cancer migration from the literature and the current study.

Author Response

Dear, reviewer 1

First of all, we deeply appreciate your critical and helpful comments. Your comments have been of great help to our current and future research. Herein, we will try to reply your comments as much as possible. Again, thank you for your comments.

Reviewer 2 Report

In the manuscript entitled “AMPKα1 regulates cancer progression by regulating TLR4-medicated TRAF6-BECN1 signaling axis”, Kim et al. described the role of AMPKa1 in regulating cancer progression and TRAF6-BECN1 autophagic signaling mediated by TLR4.

In general, data are sound and the employed methodologies are quite appropriate.

The work have some weaknesses, starting from the explanation of the novelty and of the objectives of this study, that prevent this manuscript from being recommended for publication in this version. It is extensively known how AMPK contributes at the autophagy response, by modulating BECN1 activity through its phosphorylation, and its role in cancer cells migration. Moreover, I’d suggest to introduce and discuss on translational outcomes of this work and the impact on cancer.

The overall data are not viewed to be sufficient to strongly support the conclusions described in the manuscript. Differently from what the title introduces, the authors speculated about the possibility that AMPKa1 plays a pivotal role in cancer progression by regulating the TRAF6-BECN1 signaling axis for autophagy induction, but no experiments were carried out in cancer models to demonstrate they are in the same complex, as suggested, and they are activated/regulated to induce autophagy in response to proper stimuli.

The authors should better evaluate the mechanism in a specific cancer context, I’d suggest lung cancer, as the work has been mainly described in lung models (cancer cell lines and patients). Following what Zhan et al published (Autophagy facilitates TLR4 and TLR3- triggered migration and invation of lung cancer cells through the promotion of TRAF6 ubiquitination, Autophagy 2013), TLR4-induced autophagy promotes invasion of lung cancer cells. In particular, they showed that lung cancer cells express TLR4, which is stimulated by LPS and triggers autophagy by inducing the conversion of LC3-I in LC3-II and increases the expression of BECN1 and degradation of SQSTM1. In order to investigate whether AMPKa1 is involved in TLR-induced autophagy in lung cancer cells, AMPKa1 knockdown/overexpression effects should be compared with those obtained with TLR4 inhibition/ablation/stimulation. Their hypothesis should be demonstrate in different lung cancer cells which have active and overexpressed TLR4 and AMPKa1, providing some controls relative to TLR-induced autophagy response, other than LC3 (i.e. BECN1 expression and others).

  • AMPKa1 association with TRAF6 and BECN1 should be analyzed also in endogenous context (interactions and PTMs), specifically in lung cancer cells, in order to characterize the formation of the identified functional complex after proper stimulation.
  • Figure 3. The authors should explain why they used THP-1 cells to assess AMPKa1 involvement in the autophagic flux, as already suggested same analysis should be translated in cancer context.
  • TLR4 (or its downstream targets) inhibitors (i.e. TAK-242) should be used for assays in which the authors analyze TLR4-induced signaling axis (Figs. 3, 4, 6)
  • Migration and invasion assays should be associated with the analysis of the TLR4 pathway activation and therefore BECN1/TRAF6 involvement.
  • To evaluate cancer cells invasion ability, the authors should perform also a matrigel invasion assay.
  • The authors should examine whether the autophagy induction mediated by AMPKa1 impacts on production of proinflammatory and immunosuppressive cytokines, chemokines, and MMPs triggered by TLR4 in lung cancer cells (with LPS and 3MA treatment).
  • In general, many experiment are lacking details. Please, include experimental information, (i.e. Immunofluorescence analysis: DAPI usage)
  • HCMDB and GEPIA data analysis should be better described.

Fig7A-B: please indicate which values are indicated on the y-axis. Number of patients for each group should be provided. Please also explain which statistical test was used to compare Group A and B.

Minor comments:

  • Graphic representation of results obtained in Fig 2F and analyzed with imageJ should be provided
  • Typing errors should be checked (i.e. see Title and line 295)
  • Figure 4, 6: statistical analysis between different treatment conditions on the same cell line, other than between the two different cell lines, should be provided.
  • Figure 6A, C, D, F: LPS + 3MA and LPS + CQ data lack of statistical analysis, which should be provided as for vehicle and LPS.

Author Response

Dear, reviewer 2

First of all, we deeply appreciate your critical and helpful comments. Your comments have been of great help to our current and future research. Herein, we will try to reply your comments as much as possible. Again, thank you for your comments.

Round 2

Reviewer 1 Report

Authors have answered most of the questions.

Minor comments:

  1. On page 9, in the revised Fig. 2A, the left part is redundant and suggested to be removed. Since the antibody against phospho-BECN1 at T388 was not commercially available, the authors didn’t measure the T388 phosphorylation levels of Beclin 1 in Ctrl and AMPKα1KO HEK293T cells upon serum starvation in Fig. 2E. Therefore, the “T388-P” in Figure 2A is suggested to be replaced with “phosphorylation”, due to the lack of evidence to show whether AMPKα1 phosphorylates Beclin 1 at T388 upon serum starvation in the current study. In addition, “starvation” in the whole manuscript should be changed to “serum starvation”.

  1. On page 10, although it is difficult to suggest the mechanism by which AMPK affects individual gene expression, the authors could mention the findings that LPS even downregulates eight ATG-related genes in AMPKα1 KD THP-1 cells, and when gene expression levels were compared between AMPKα1KD THP-1 cells treated with LPS and Ctrl THP-1 cells treated with LPS, seven genes were even up-regulated in AMPKα1KD THP-1 cells (Fig. 3D, green bars in column 3) through undefined mechanisms in the discussion. There were similar issues for Fig. 3E.

  1. On page 11 and 12, since LPS-induced LC3-II was significantly attenuated in AMPKα1 KO A549 cells, why did treatment with CQ, an autophagy inhibitor that blocks the binding of autophagosomes to lysosomes, still induce increases of LC3-II in AMPKα1 KO A549 cells to similar levels of LC3-II in Ctrl A549 cells (Fig. 3B and 3C)?

  1. On page 19, please add specific descriptions about the role of AMPK in the regulation of cancer migration from the literature to replace “According to previous reports,32,33”.

Reviewer 2 Report

There are some points of the manuscript through which the authors would like to demonstrate that AMPKa1 regulates lung and breast cancer progression by regulating TLR4-mediated TRAF6-BECN1 signaling axis. While the interaction has been now shown even in A549 cancer cells (indeed results showed in the letter should be included in main figures), confirming their first analysis in vitro and in HEK293T cells, data used to demonstrate that cancer progression is regulated by AMPKa1 via TLR4-TRAF6-BECN1 still have same weaknesses as before. Migration and invasion assays in cancer cell lines should be associated with the analysis of the TLR4 pathway activation not only by analyzing LC3 cleavage, but even by checking the involvement of BECN1/TRAF6. Thus, I still suggest repeating the experiments with some important controls.

They should analyze AMPKa1 knockdown/overexpression effects on autophagy, migration and invasion comparing them with those obtained with TLR4 inhibition/ablation. BECN1 increased expression and activation (phosphorylation, ubiquitination…) should be showed in order to confirm that they are specifically involved in the mechanism. Data showed in HEK293T and THP-1 cells, Figure 2 and 3, supporting their hypothesis, should be demonstrated in cancer cells, at least ones used in this study.

Results showed in patient-derived NSCLCs do not dissect the molecular mechanism in cancer context, while suggest the positive association between AMPKa1 expression and autophagy flux-related genes.

Figure 4, 6 still lack of some statistical analyses (particularly on LSP+CQ results)

FIG 7A-B: is the number of patient indicated in the figure legend specific for each group A and B, or it is the total(A+B)? Please specify in order to understand whether both groups have equal or at least comparable numbers of patients.

FIG 7A-B: even if you indicate the statistical significance between the two groups at the top of each graph, the authors have still not mentioned which specific test they used to obtain the indicated p-values.

The authors should better describe HCMDB and GEPIA data analysis process (materials and methods section).

Round 3

Reviewer 2 Report

FIG 7A-B: On HCMDB patients with primary tumors with metastasis are only 2, while 166 samples have primary tumor without metastasis (Exp ID, EXP00003). With this samples, the analysis is considered unpresentable, it is better to remove this data from the manuscript. The only way to present this comparison is to expand the number of patients, by considering more than one Exp IDs of the dataset, whether possible, in order to analyze two groups with a reasonably comparable number of patients. At the same time, for Figure 7B, they should mention that Group A has 4 samples, while Group B has 8 (Exp ID, EXP00006). Number of patients for each group should be always specified.

I’d still suggest adding some important controls to the data presented, with the aim of demonstrating that AMPKa1 regulates cancer progression through the specific TRAF6-BECN1 signaling axis in cancer cells, as the title explains. However, considering author’s answers and discussion, the manuscript could now be accepted anyway.
